# Impacts of Defocusing Amount and Molten Pool Boundaries on Mechanical Properties and Microstructure of Selective Laser Melted AlSi10Mg

**DOI:** 10.3390/ma12010073

**Published:** 2018-12-26

**Authors:** Suyuan Zhou, Yang Su, Rui Gu, Zhenyu Wang, Yinghao Zhou, Qian Ma, Ming Yan

**Affiliations:** 1Department of Materials Science and Engineering Shenzhen Key Laboratory for Additive Manufacturing of High-Performance Materials, South University of Science and Technology of China, Shenzhen 518055, China; zhousy@sustc.edu.cn (S.Z.); 2161160220@email.szu.edu.cn (Y.S.); gurn@mail.sustc.edu.cn (R.G.); wangzy7@mail.sustc.edu.cn (Z.W.); 11649022@mail.sustc.edu.cn (Y.Z.); 2Key Laboratory of Artificial Micro- and Nano-Materials of Ministry of Education and School of Physics and Technology, Wuhan University, Wuhan 430072, China; 3Centre for Additive Manufacture, School of Engineering, RMIT University, Melbourne, VIC 3000, Australia; ma.qian@rmit.edu

**Keywords:** aluminum alloys, selective laser melting (SLM), mechanical properties

## Abstract

The influences of processing parameters such as volumetric energy density (ε) and, particularly, defocusing amount (DA) on densification, microstructure, tensile property, and hardness of the as-printed dense AlSi10Mg alloy by selective laser melting (SLM) were studied systematically. The molten pool boundaries (MPBs) were found overwhelmingly at regular and complex spatial topological structures affected by DA value to exist in two forms, while the “layer–layer” MPB overlay mutually and the “track–track” MPBs intersect to form acute angles with each other. The microstructure of MPBs exhibits a coarse grain zone near the MPBs and the characteristics of segregation of nonmetallic elements (O, Si) where the crack easily happened. The DA value (−2 to 2 mm) affected both the density and the tensile mechanical properties. High tensile strength (456 ± 14 MPa) and good tensile ductility (9.5 ± 1.4%) were achieved in the as-printed condition corresponding to DA = 0.5 mm. The tensile fracture surface features were analyzed and correlated to the influence of the DA values.

## 1. Introduction

Additive layer manufacturing (ALM) technology as a layer-additive approach has been widely used and developed for more than 30 years [1,2]. Selective Laser Melting (SLM) is an ALM technique to produce Fe-, Ti-, Ni-, and Al-based alloys by traces of one cross section of the CAD-data [3,4]. The near-net-shaped and near-fully-dense parts can be made by building on the successive layers by using fast-moving laser beam as illustrated in Figure 1a, where a high intensity laser scanned the powder bed selectively layer by layer. The metal powder particles are melted and form molten pool, which rapidly solidifies during the short-term non-equilibrium phase transition. Bandar AlMangour et al. (2018) pointed out that the volumetric laser energy influenced the microstructures and porosity of SLMed TiC/316L nanocomposite parts [5,6]. Wu et al. (2016) revealed the SLM process offers a wide range of advantages including significant microstructural refinement by small grain sizes as compared to other conventional processes, due to the intrinsic high thermal gradients of the SLM process as discussed [7]. Murr et al. (2009) found that the yield and tensile strength of dense SLMed Ti6Al4V alloy increased by nearly 50% greater than forgings [8]. Michele et al. (2016) revealed the SLMed metallurgy of high-silicon steel can be changed from <001> fibre-texture to a cube-texture at higher energy input [9]. 

Al-Si based traditional light-weight, castable alloys have been widely used in aerospace and automotive industries because of their good strength, low density, and good castability. Li et al. (2016) found SLMed NTD-Al have strong interfacial bonding in the Al matrix with rod-like nano-Si and TiB_2_ particles due to solidification of non-equilibrium and rapid cooling [10]. Prashanth et al. (2017) observed the supersaturated Al solid solution of the SLM-processed samples with fibrous eutectic Al/Si has good weldability [11]. Ma et al. (2016) pointed out the SLMed Al-20Si alloy with eutectic Al/Si at high pressure solidification can improve the mechanical properties [12]. Read et al. (2015) acquired near-fully-dense SLMed AlSi10Mg parts by suitable process parameters optimization resulting in good mechanical properties [13]. Aboulkhair et al. (2014) reduced porosity in AlSi10Mg parts by used the different parameters and scan strategies [14]. Brandl et al. (2012) found the fatigue life of AlSi10Mg samples at different directions can be changed at a peak-hardened (T6) state [15], Amato et al. (2012) observed that the yield strength and UTS of SLMed Inconel 718 along the horizontal direction was higher than in the vertical plane mainly due to the difference in the crystallography and microstructure [16,17]. Thijs et al. (2013) showed that the SLMed AlSi10Mg alloy has a different texture, using rotation of 90° of the scanning vectors [17]. Abe et al. (2003) found that the low density and high porosity of the SLMed samples lead to the low yield tensile strength [18]. Li et al. (2012) found that the selective laser melting Al12Si alloy has 25% tensile ductility refined by eutectic microstructure after solution heat treatment [19]. Rosenthal et al. (2017) found that the SLMed AlSi10Mg alloy has a strain rate that is sensitive to changes in strain hardening and flow stress [20]. 

To maximize the potential of the SLM process for this alloy and other similar metallic materials, most of the relevant studies have focused on processing parameters optimization, e.g., energy density, scanning strategy, hatch space, to explain the appearance of micropores, microstructures, crystal texture, and internal residual stress. Khairallah et al. (2012) showed that the melt flow has three sections such as a topological depression, a transition, and a tail region to generate pore defects [21]. However, limited attention has been addressed to the influence of two other important factors, namely the molten pool boundaries (MPBS) and the defocusing amount (DA). Some studies have proposed the existence of MPBs in the as-built SLMed parts because the rapid high heat input by a moving laser beam repeats the deposition layer after layer results in overlapping of molten pools. Lin et al. (2006) showed the directional heat transfer along the molten pools relative to the bottom has a lower solidification rate [22]. Yadroitsev et al. (2014) observed the MPBs existed clearly and interconnected to form a specific spatial topological structure in the Ti6Al4V SLM parts [23]. Wen et al. (2014) discussed how the performance of SLM parts was greatly affected by the whole MPBs structure in different building directions, because the properties of individual MPBs easily form plane grains which differ from that of other regions [24]. 

The defocusing amount (DA) is defined as the distance that the laser radiation penetrated into the powder layer and up to a certain distance from the surface. It is recommended as a critical factor for the SLM process because the overlapping of multi-tracks or multi-layers during SLM leads to MPBs with a special shape in the microstructure as illustrated in Figure 1b. To the authors’ best knowledge, the influence of either factor on the mechanical properties and density of the SLMed AlSi10Mg has not been investigated in detail yet. This study is therefore carried out to fill this knowledge gap. The research findings highlight the importance of both factors in affecting the mechanical performance and microstructure of the SLMed AlSi10Mg alloy.

## 2. Experimental

Commercial as-purchased AlSi10Mg powder was used (AMC powders GmbH, Beijing, China). The chemical component of the AlSi10Mg powder is listed in Table 1. The size ranges from 20–60 μm, as measured by a laser diffraction particle analyzer Coulter LS230 (manufacturer, city, country). The SLM process is based on a Concept Laser M2 (Concept Laser GmbH, Lichtenfels, Germany) with a build chamber of 280 mm × 250 mm × 250 mm. The build chamber was filled with high-purity argon and the oxygen content inside of the chamber was below 15 ppm. The operating maximum power of the IPG fiber laser was 400 W. The thickness (30 μm) of powder layer and hatch spacing (100 μm) were kept constant during the SLM processing. The laser processing parameters such as energy density and DA were optimized. Sixty cylindrical samples and tensile samples were produced with scan speed (*v*) at 1400 mm/s and a varying laser power (*P*) in order to obtain the highest density as shown in Table 2. The laser beam is raster-scanned individually with small islands in order to reduce the residual stresses as for the laser scanning mode in Figure 1c [17]. The island size was about 5 × 5 mm^2^ being built continuously and randomly and the scanning vectors shift 1 mm in both x and y directions. The defocusing amount can be controlled by manipulating the movement of the sample platform up and down via using the laser displacement sensor (Figure 1b). The displacement accuracy is ±0.01 mm.

In this work, specimens to characterize the fractography were polished and finally etched in a solution consisting of distilled water (90 mL) with HF (10 mL) for 25 s. The macro- and microstructures of the tensile specimens were analyzed by a Zeiss Axioscop 40 Pol polarising microscope (Zeiss, Oberkochen, Germany) and scanning electron microscope (SEM) at 20 kV (Quanta 200, FEI Company, Eindhoven, the Netherlands). Chemical compositions of the crystalline phases in the as-printed alloy were investigated by an energy dispersive X-ray (EDX) spectroscope (EDAX Inc., Mahwah, NJ, USA). Phase identification was observed by a D8 Advanced X-ray diffractometer (XRD) with Cu Kα radiation (λ = 0.15418 nm) (Bruker AXS GmbH, Karlsruhe, Germany) at 20 kV and 20 mA using a continuous scan mode at 2°/min, The micro-CT experiments were analyzed at the high-resolution comprehensive microfocus CT detection system diondo d2 at 100 kV with the best resolution at ~2 μm. The crystallographic orientations of the samples were investigated via Electron Backscattered Diffraction (EBSD) by an SEM (ZEISS Merlin, Oberkochen, Germany) with a EDAX/Digiview 4 system. The relative density was determined by the Archimedes method as a percentage of the material’s bulk density of 2.68 g/cm^3^.

The crosshead speed of 1 mm/min was used in the tensile tests with an Instron 25 mm dynamic extensometer (Norwood, MA, USA) for controlled displacement. Microhardness tests were conducted with five repetitions per sample using a Vickers hardness tester FV-700 (FUTURE-TECH, Kawasaki, Japan) with 100 gf load for 15 s. 

## 3. Results and Discussion

### 3.1. Crystalline Phases in the as-Printed AlSi10Mg

Figure 2a shows the SEM results of the spherical powder of an average size of 42 μm. Figure 2b shows two dominant XRD peaks belong to face-centered cubic (fcc) α-Al and eutectic Si-particles. The presence of the Mg_2_Si precipitates is also detected, although the corresponding XRD peak is rather weak. The phases revealed by the XRD are typical of the as-printed AlSi10Mg alloy.

### 3.2. Effect of Laser Energy Density and Defocusing on Densification 

In Figure 3a,b, the relative density of the as-printed AlSi10Mg vs. the “volumetric energy density” (ɛ) and DA, is shown, respectively. The *ɛ* is defined as follows:$$ɛ=\frac{P}{\nu hd}$$
where *P* is the laser powder, *v* is the scanning rate, *h* is the layer thickness, and *d* is the layer width used.

In total, 60 samples parallel to the substrate were built and the SLM processing parameters are shown in Table 2. Optical micrographs of the samples are shown in Figure 3a for demonstrative purposes of the samples with low and high density. The general trend revealed by Figure 3a is that increase of *ɛ* (i.e., using a higher laser power at lower scan speeds) can lead to better densification. Figure 4a–c shows the porosity of as-printed AlSi10Mg samples from the micro-CT reconstructed profile at ε = 61.9 J/mm^3^, 76.2 J/mm^3^, 88.1 J/mm^3^ respectively. The main reason for the interior porosity formation was that insufficient energy gave unmelted spots, forming voids between the melt pool line and hatch pattern. On the other hand, there are large cavities, a ‘‘balling’’ effect, and irregular long channels caused by highly dynamic and unstable melt pool tracks at an excess of laser energy [25]. 

Figure 3b indicates that the DA has a prominent influence on density and increasing the DA to about 0.5 mm has resulted in the maximal density (via the sample of P3D6). In contrast, there is a large amount of pores existing in the microstructure if a negative DA is used, while the pulsed laser recoil force influenced by DA value has changed greatly from the micro-CT images, Figure 4d–f. If using a CW laser at negative DA, under certain conditions the so-called ‘‘keyhole-mode’’ laser welding is motivated [26,27], which forces melt pool oscillations and flow instability by disturbances from thermocapillary convection and the pulsed laser recoil force, and its depth undergoes severe oscillations, some bubbles from the metallic evaporation are trapped at the solidifying front of the tracks, leading to more near-spherical defects [28,29,30]. 

The surface roughness is reduced as shown in Figure 5a–d. Andrew Townsend et al. (2010) checked the roughness of additive manufactured (AM) Ti6Al4V samples used X-ray computed tomography (CT) [31]. The melt pool is forced to oscillations and flow instability by disturbances from thermocapillary convection and the pulsed laser recoil force, which results in a rough surface as shown in Figure 5a,b. The temperature gradient are intensified and in turn created surface tension gradient caused by the resultant turbulence of Marangoni flow within the molten pool, with a further increase in the laser energy, the melting pool size becomes bigger and the flow inside would be stabilized to eliminate this type of pores, followed by smoother surface roughness as shown in Figure 5c,d. For different DA values, there is no remarkable influence on the surface roughness because the volumetric energy density was a major influence factor for stability of the molten pool [32].

### 3.3. MPBs in the as-Printed AlSi10Mg 

The cross section of a melt pool formed during SLM can be described approximately as an arched structure schematically shown in Figure 6a. By means of the metallography, two types of MPBs can be found in the SLMed samples. One is the arched “layer–layer” parallel type of MPBs (Figure 6b) and the other is the “track–track” overlapping type of MPBs (Figure 6c). It is found that the parallel “layer–layer” MPBs connect with short “track–track” MPBs and form acute angles. Aside from sample density, the DA value also significantly affects the shapes of the MPBs (Figure 6d).

When DA value is negative (i.e., −2 mm < DA < 0 mm), as illustrated in Figure 1b, the laser melting seems to lead to the formation of a deep penetration in the powder bed, and the corresponding MPBs are closer to a cone-shaped “layer–layer” MPBs heaped up and “track–track” MPBs adjacently intersected with each other to form acute angles (0° < θ_T_ < 45°). In the case of DA = 0, the MPBs cross-section of a melt pool has small ‘‘fish scale’’ morphology. When the DA increases from 0 mm to 2 mm, the plasma is more likely to be generated and the laser melting mode transits from the so called ‘keyhole mode’ to the ‘conduction mode’ as known in laser welding. In this case, the MPBs are prone to be the arched type (Figure 5d). It is found that the “track–track” MPBs usually intersect and form big angles (45° < θ_T_ < 90°) while arcuate “layer–layer” MPBs are heaped up almost in parallel. 

The slipping shear stress *ψ* of the horizontal tensile specimens along the slipping direction can be expressed as follows [24] (1):(1)$$\psi =\frac{F}{S}\cos\theta \cos\lambda$$

In Equation (1), *S* is the cross section, which the tensile load *F* is applied to, θ is the angle between the tensile load and the slip direction, and λ is the angle between the tensile load and the normal direction of the slip plane. The critical shear stresses $\psi_{k}$ up to the yield point σs when the tensile stress (*F**/**S*) increases to a critical value. The yield limit σs and the critical shear stress $\psi_{k}$ are expressed as follows:(2)$$\psi_{k}=\sigma s\cos\theta \cos\lambda$$
(3)$$\;\sigma s=\psi_{k}\frac{1}{\cos\theta \cos\lambda }$$

When the loading force *F* of the horizontal samples is parallel with the X–Y plan, the angle λ between the loading force *F* and the slipping direction is near 0°. The critical shear stress $\psi_{k}$ is only connected with the interfacial binding force of the slipping surface regardless of the change of the applied loads *F*, therefore the yield limit σs depends only on the variation of the angle *θ.*

Figure 7a shows the SEM images of two kinds of molten pool boundaries (MPBs): “layer–layer” and “track–track” MPBs on the cross section. The MPBs shown a honeycomb appearance and the original dendrite arm, which intersect each other to form acute angles shown in Figure 7b. The growth rate (R, m/s) and thermal gradient (G, K/m) during the SLM processing determine the microstructure of samples, when the thermal gradient is perpendicular to the laser scan movement direction, solidification mode varies at the melt pool boundaries (MPBs) edge leading to the coarse grain zone with roughly the width of 2 μm, which was similar with the weld heat-affected zone in Figure 7c. The energy spectrum along a line (AB) shown the nonmetallic elements (O, Si) cross the coarse grain zone perpendicular to the MPBs as shown in Figure 7d, where cracks are easily initiated and have a deteriorative effect on mechanical properties of the SLM samples.

The cross-section microstructure of the as-printed specimen along the building direction is shown by the EBSD image (Figure 8a), where columnar grains with the size of 10–20 μm and intergranular spacing are generated. The orientation difference distribution of adjacent grains shown in Figure 8b, the misorientation angle about 35°–60°, accounted for most of the large angle grain boundary, which makes growth and propagation of cracks difficult, and consequently increases the toughness of the material. 

### 3.4. Tensile Mechanical Properties 

Table 2 summarizes the detailed tensile properties obtained from the thirteen batches of the SLMed AlSi10Mg alloy samples printed under different DA values (see Table 2) along with their micro-hardness results, provided by ASTM Standard B26/B26M. The best tensile properties achieved are tensile strengths of ~456 ± 14 MPa and elongation of 9.5 ± 1.4%, corresponding to the DA of 0.5 mm. Figure 9a provides representative tensile stress–strain curves of the as-printed AlSi10Mg alloy. It is noted that the variations of the DA have resulted in markedly different tensile properties. 

Besides grain slipping, The SLMed parts slip along the MPBs form ductile deformation preferentially because of poor adhesion and low stability in the junction between the MPBs. The tensile fracture surfaces display distinctly different features due to different DA values. When the DA increases from 0 mm to 2 mm, the cross section of horizontal processing specimens shows a particular wavy and serrated morphology, related to the surface of MPBs comprising of a mixture of arched parallel “layer–layer” and “track–track” overlapping MPBs, as shown in Figure 9b,c, which is supposed to be a contributing factor to the combination of high strength and good ductility. When DA value is negative (i.e., −2 mm < DA < 0 mm), the crack surfaces appear smoother than those of the other tensile samples. The fracture surfaces contain a mixture of smooth areas with porosity or shrinkage cavities and small dimples as shown in Figure 9d. Cleavage step and non-melted particles are also observed in Figure 9b–d; furthermore, the equiaxed submicrometer dimples with the same hot crack are observed at the whole fracture surfaces shown in Figure 10a,b.

Regardless of how the DA values changes when increase from −2 mm to 0 mm, the sliding surfaces containing “layer–layer” MPBs are almost planes and always vertical to the tensile loading (θ_L_ = 0°) for the horizontal specimens (Figure 6c). It is difficult along the arched parallel “layer–layer” MPBs slipping surfaces and mainly to slip along overlapping “track-track” MPBs surfaces with the continuous increase of the tensile load (0° < θ_T_ < 45°) until the occurrence of fracture. Such cracks extend along the “layer–layer” MPBs and the formation of cleavage surfaces occurs, indicating a brittle fracture feature.

When the DA increases from 0 mm to 2 mm, the sliding surfaces consisting of the actual “layer–layer” MPBs surfaces are not regular planes and arched shape “layer–layer” MPBs are also not vertical to the tensile loading (45° < θ_L_ < 90°). On the other hand, the ductile deformation is easily formed by the slipping along “track–track” MPBs surfaces. Therefore, the tensile fracture SLM samples achieve the best ductility and highest elongation as Equation (3) due to the slipping alone both “track–track” (45° < θ_T_ < 90°) and “layer–layer” MPBs (45° < θ_L_ < 90°). With a combination of appropriate DA values, the as-printed AlSi10Mg can show good facture strength as well as good ductility, revealing a ductile fracture mechanism. The P3D6 sample is such a typical example. In summary, the DA has significant impacts on the fracture mode, macroscopic plastic behavior, and microscopic slipping of the as-printed alloy by influencing the shape of MPBs in SLM process.

## 4. Conclusions

In this study, the crack-free and near fully dense AlSi10Mg samples were fabricated using optimized SLM parameters. The processing-microstructure-mechanical properties correlation was established, allowing us to draw the following conclusions:

(1) It has been shown that the densification behavior of SLMed AlSi10Mg parts was significantly affected by the volumetric energy density (ε) and, particularly, defocusing amount (DA) during SLM. the minimum porosity and the relatively high relative densities of over 98.5% are possible at the critical energy density of ~76.2 J/mm^3^ at high laser power of about 320 w and scan speed of 1400 mm/s. With further increase of the laser energy, the melting pool size became bigger and the flow inside would to be stabilized to eliminate this type of pores and then smoother surface roughness. Simultaneously, the combination of ε and appropriate DA values of about 0.5 mm ensures the highest relative densities of over 99% for this alloy.

(2) The multi-layer and multi-track molten pools track overlapped to generate “layer–layer” and “track–track” MPBs, which have a significant impact on the plastic deformation behavior of the as-printed alloy by the slip theory analysis, the coarse grain zone can be found below the MPBs containing the segregation of nonmetallic elements where cracks are initiated easily. 

(3) The defocusing amount (DA) is one of the main factors except for energy density that affects both MPBs’ spatial topological structure as well as the density and the mechanical properties of the as-printed AlSi10Mg. The tensile fracture surfaces display distinctly different features due to different DA values. The tensile fracture SLM samples achieve the best ductility and highest elongation due to the slipping along both “track–track” (45° < θ_T_ < 90°) and “layer–layer” MPBs (45° < θ_L_ < 90°) when the DA increases from 0 mm to 2 mm. A ductile, dimpled failure mode along the two interlaced distribution of types of MPBs is observed in these as-printed specimens. High tensile strength (456 ± 14 MPa) and good tensile ductility (9.5 ± 1.4%) and microhardness of 122 ± 2 HV0.1 were achieved in the as-printed condition corresponding to DA = 0.5 mm. 

## Figures and Tables

**Figure 1 materials-12-00073-f001:**
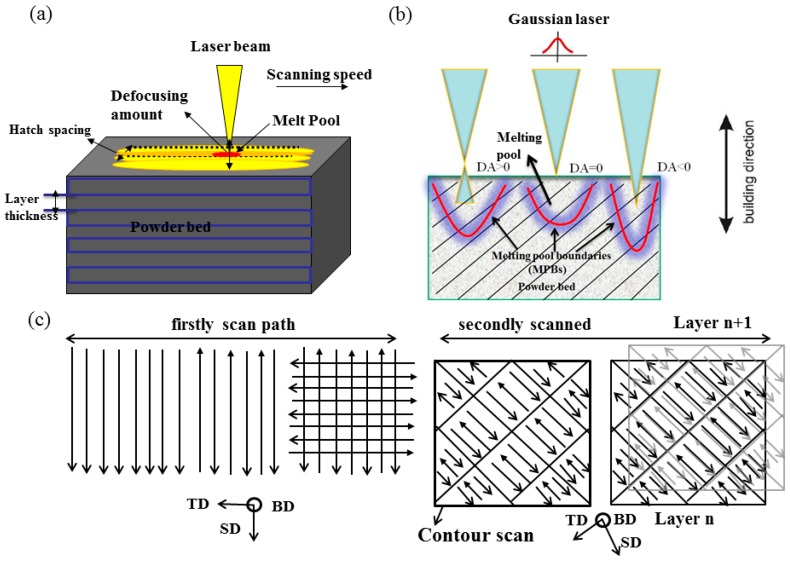
Schematic diagram of Selective Laser Melting (SLM) (**a**); Schematic illustration of the melt pools produced in different defocusing amounts (DAs) with variation of shape and discontinuity (**b**); the scan strategy “the island” (**c**).

**Figure 2 materials-12-00073-f002:**
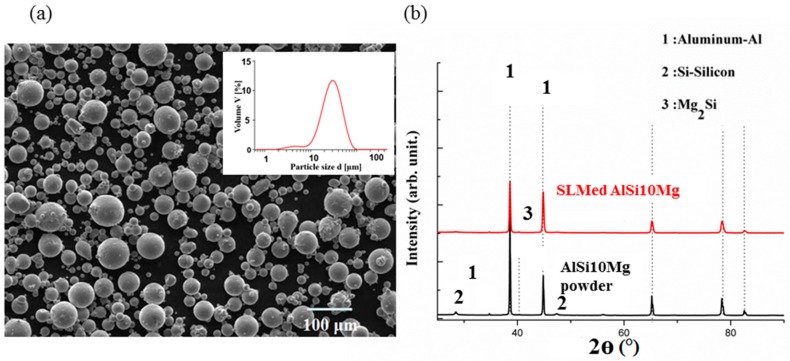
SEM images of the AlSi10Mg spherical powder and size distribution (**a**); Diffraction patterns of powdered samples in as-built conditions (**b**).

**Figure 3 materials-12-00073-f003:**
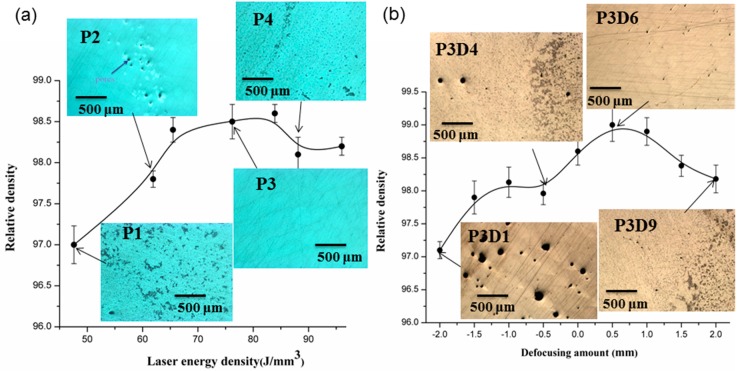
Relative density curves plotted against laser energy density and DA values and optical microscopy (OM) images of polished SLM-processed specimens (**a**,**b**).

**Figure 4 materials-12-00073-f004:**
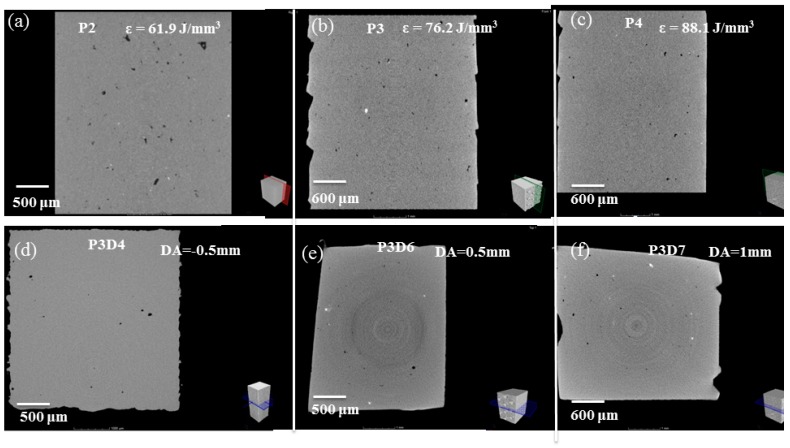
Tomographic images of SLM-processed AlSi10Mg samples at volumetric energy density (ɛ) about, 61.9 J/mm^3^ (**a**), 76.2 J/mm^3^ (**b**), 88.1 J/mm^3^ (**c**), and DA values about, (**d**): −0.5 mm; (**e**): 0.5 mm; (**f**): 1 mm.

**Figure 5 materials-12-00073-f005:**
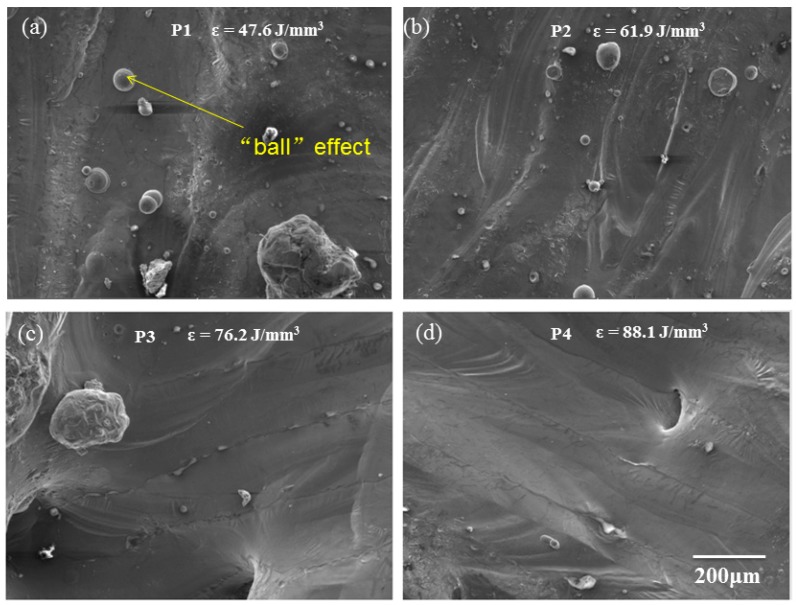
SEM images of the top surface of SLM-processed AlSi_10_Mg samples P1–P4 (**a**–**d**), all images are at the same magnification.

**Figure 6 materials-12-00073-f006:**
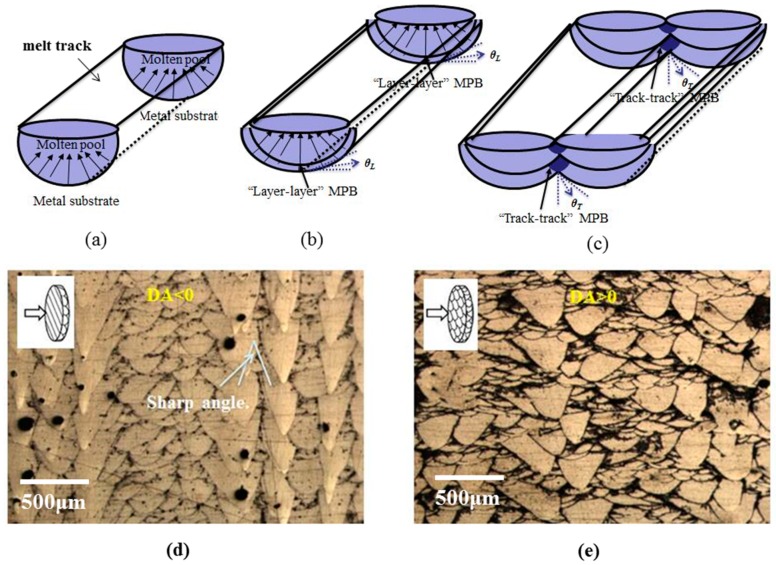
Schematic overview of the solidification of molten pools during the SLM process: (**a**) single half-cylindrical molten pool; (**b**) “layer–layer” molten pool; (**c**) “track–track” molten pool. The overlaying of molten pools with differently shaped molten pool boundaries (MPBs) of ‘‘fish scale’’ morphology produced in different DAs; (**d**) DA < 0; (**e**) DA > 0, respectively.

**Figure 7 materials-12-00073-f007:**
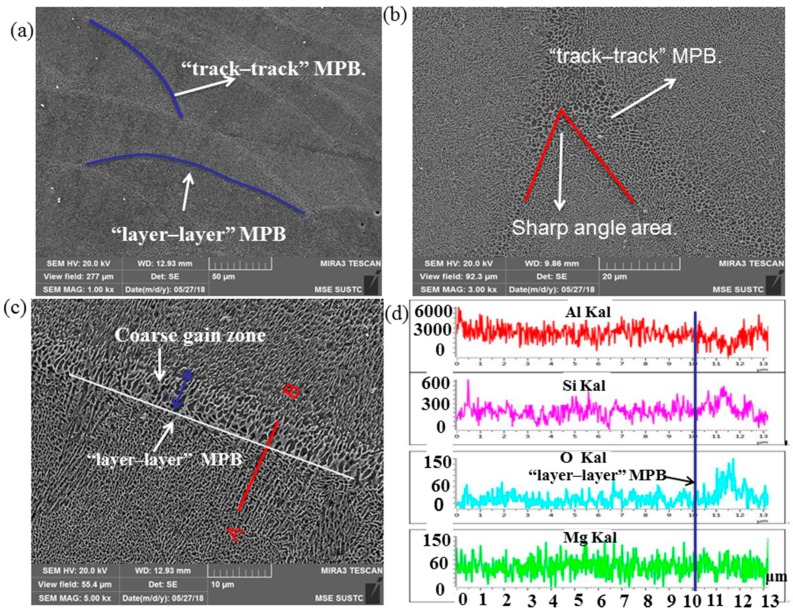
(**a**) SEM images of the MPBs in SLM samples in a section plane: (**a**) the cross section of low magnification morphology of the MPBs at the build direction; (**b**) SEM image of the morphology of “track–track” MPBs; (**c**) high magnification of morphologies of “layer–layer” MPBs; (**d**) the distribution of elements near MPB on the A–B line.

**Figure 8 materials-12-00073-f008:**
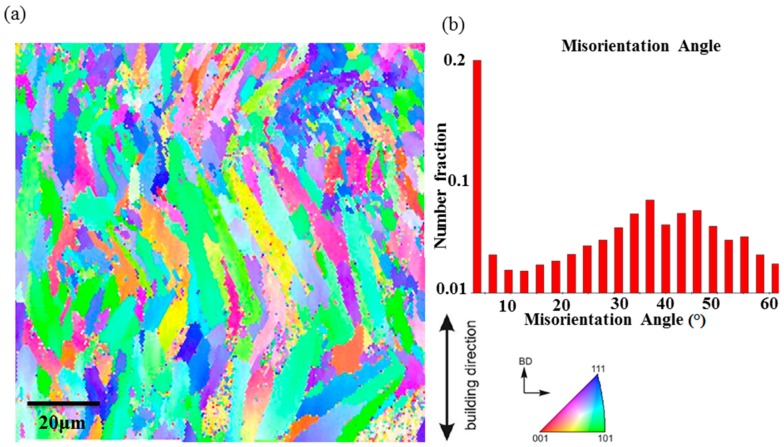
(**a**) The Electron Backscattered Diffraction (EBSD) orientation map of as-built AlSi10Mg sample at the building direction (BD); (**b**) Misorientation angle distribution.

**Figure 9 materials-12-00073-f009:**
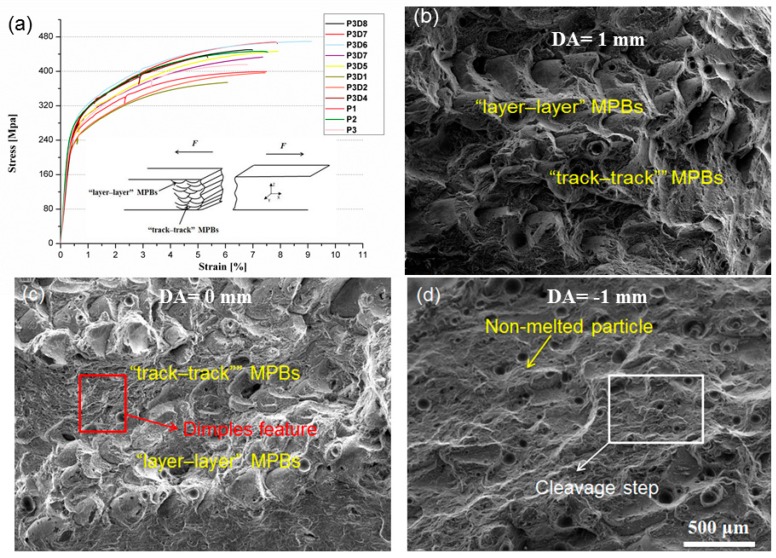
Engineering stress–strain curves of the as-built samples of horizontal direction at different laser power and DA, the Illustrations is the stress cross sections of the horizontal SLMed samples (**a**); the SEM image shows the fractured surface of SLM AlSi10Mg samples with the characteristic microstructure at laser power of P = 320 W and fixed scanning speed of 1400 mm/s but various DA of (**b**): 1 mm; (**c**): 0 mm; (**d**): −1 mm.

**Figure 10 materials-12-00073-f010:**
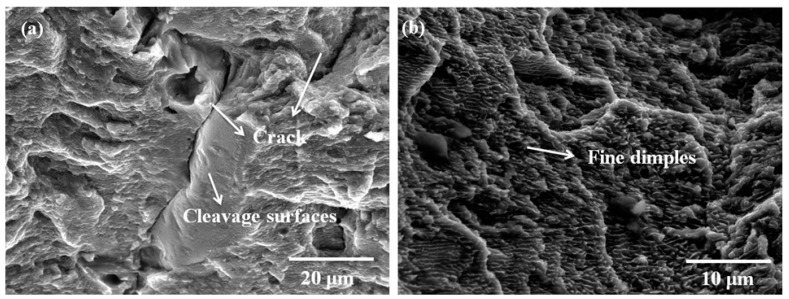
The high magnification SEM images shown fracture surface with crack (**a**) and mixture of small dimples (**b**).

**Table 1 materials-12-00073-t001:** Chemical composition of the AlSi10Mg powder.

Weight (%)	Si	Fe	Zn	Mn	Mg	Ni	O	Cu	Al
AlSi10Mg	9.0–11.0	≤0.55	≤0.10	≤0.45	0.2–0.45	≤0.05	0.02	0.03	balance

**Table 2 materials-12-00073-t002:** Parameters for optimizing the density of the as-printed AlSi10Mg and the mechanical properties ultimate tensile strength (UTS), yield strength (YS), and elongation of AlSi10Mg fabricated by SLM at different DAs.

Batch Number	Power (W)	Scan Speed (mm/s)	Defocusing Amount DA	Ultimate Tensile Strength (MPa)	Yield Strength (MPa)	Microhardness (HV)	Young Modulus (GPa)	Elongation (%)
P1	200	1400	0	390 ± 16	170 ± 3	112 ± 10	62.1	6.5 ± 1.3
P2	260	1400	0	385 ± 18	165 ± 2	114 ± 6	63.7	6.8 ± 1.6
P3	320	1400	0	429 ± 20	175 ± 6	121 ± 11	66.9	7.5 ± 1.5
P4	370	1400	0	415 ± 12	178 ± 8	124 ± 8	70.6	6.9 ± 1.4
P3D1	320	1400	−2	370 ± 10	160 ± 6	117 ± 9	65.4	6.1 ± 1.3
P3D2	320	1400	−1.5	395 ± 12	177 ± 6	119 ± 6	63.7	7.3 ± 1.2
P3D3	320	1400	−1	390 ± 16	150 ± 6	120 ± 10	65.9	5.8 ± 1.3
P3D4	320	1400	−0.5	385 ± 17	182 ± 6	121 ± 10	62.3	7 ± 1.2
P3D5	320	1400	0	429 ± 20	175 ± 6	121 ± 11	66.9	7.5 ± 1.5
P3D6	320	1400	0.5	468 ± 14	191 ± 6	122 ± 12	71.2	9.5 ± 1.4
P3D7	320	1400	1	430 ± 18	182 ± 6	116 ± 12	69.5	7.8 ± 1.2
P3D8	320	1400	1.5	440 ± 15	177 ± 6	114 ± 16	63.5	8.2 ± 1.6
P3D9	320	1400	2	425 ± 16	189 ± 6	119 ± 12	68.2	7.6 ± 1.2

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
