# Peer review of "Impacts of Defocusing Amount and Molten Pool Boundaries on Mechanical Properties and Microstructure of Selective Laser Melted AlSi10Mg"

_materials, 2018, doi:10.3390/ma12010073_

Reviewer 1 Report

In this manuscript, the authors studied the Impacts of Defocusing Amount and Molten Pool
Boundaries on Mechanical Properties and Microstructure of Selective Laser Melted AlSi10Mg. The paper in general is interesting but require further modifications:

1- Figure 1C is completely copied from previous published reference. This is plagiarism!

2- Data in Table 2 looks like it is not from a real experiment. Please verify !

3- How "defocusing amount" has been measured and controlled?

4- There are a great deal of recently published papers concerning thermal behavior during SLM processing and its effect on the microstructure and mechanical properties. The authors need to cite the following example papers and link their findings with previously published work:

- Selective laser melting of nano-TiB2 decorated AlSi10Mg alloy with high fracture strength and ductility

- Densification behavior, microstructural evolution, and mechanical properties of TiC/316L stainless steel nanocomposites fabricated by selective laser melting

- Thermal behavior of the molten pool, microstructural evolution, and tribological performance during selective laser melting of TiC/316L stainless steel nanocomposites: Experimental and simulation methods

- Strain rate sensitivity and fracture mechanism of AlSi10Mg parts produced by selective laser melting

Author Response

Reviewer #1:

Comment 1: 1- Figure 1C is completely copied from previous published reference. This is plagiarism!

Response: We apologize for our haste and overlook. We did not intend to do this. The reference for Fig. 1C was given in the main text of the original submission but it was not mentioned clearly in the corresponding figure caption. Anyway, we have redrawn Figure 1c and cited the relevant literature correctly in the revised manuscript. Please also refer to the Table 1 (in the end of the Response Letter) for the corresponding revision.

 Comment 2: 2- Data in Table 2 looks like it is not from a real experiment. Please verify !
Response: Thanks for the suggestion. We are sorry for that we did not present the data well. The manuscript has been thoroughly checked again and the tensile curve was added according to the experimental data obtained.

 Comment 3: How "defocusing amount" has been measured and controlled?

Response: We have done the SLM processing on a Concept Laser M2 printer. Through checking the operation manual and consulting with service engineers, we got to know that the defocusing amount can be controlled by manipulating the movement of the sample platform up and down via using the laser displacement sensor (please see below Figure 1a). The displacement range is ±5 mm and accuracy is ± 0.01 mm. Aside from this, automatic focusing device was installed & employed before each printing to ensure the initial defocusing amount is 0 mm (please see below Figure 1b).

 Fig. 1 (a) Snapshot of the focus move and laser parameters page of the control software (for monitoring defocus amount) and (b) snapshot of the focus move of the control software (for monitoring defocus amount before each printing).

 Comment 4: There are a great deal of recently published papers concerning thermal behavior during SLM processing and its effect on the microstructure and mechanical properties. The authors need to cite the following example papers and link their findings with previously published work:

Selective laser melting of nano-TiB2 decorated AlSi10Mg alloy with high fracture strength and ductility

Densification behavior, microstructural evolution, and mechanical properties of TiC/316L stainless steel nanocomposites fabricated by selective laser melting

Thermal behavior of the molten pool, microstructural evolution, and tribological performance during selective laser melting of TiC/316L stainless steel nanocomposites: Experimental and simulation methods

Strain rate sensitivity and fracture mechanism of AlSi10Mg parts produced by selective laser melting

Response: Thanks for the suggestion. We have read the suggested references carefully and they have been cited in the revised manuscript. Details of the references are as follows:

X.P. Li , G. Ji , Z. Chen, A. Addad , Y. Wu , H.W. Wang, J. Vleugels, J. Van Humbeeck, J.P. Kruth, Selective laser melting of nano-TiB2 decorated AlSi10Mg alloy with high fracture strength and ductility. Acta Materialia 129 (2017) 183-193.

Bandar AlMangour, Dariusz Grzesiak, Tushar Borkar, Jenn-Ming Yang. Densification behavior, microstructural evolution, and mechanical properties of TiC/316L stainless steel nanocomposites fabricated by selective laser melting, Materials and Design 138 (2018) 119–128.

Bandar AlMangoura, Dariusz Grzesiak , Jinquan Cheng , Yavuz Ertas, Thermal behavior of the molten pool, microstructural evolution, and tribological performance during selective laser melting of TiC/316L stainless steel nanocomposites: Experimental and simulation methods. Journal of Materials Processing Tech. 257 (2018) 288–301.

I. Rosenthal, A. Stern, N. Frage, Strain rate sensitivity and fracture mechanism of AlSi10Mg parts produced by selective laser melting. Materials Science & Engineering A 682 (2017) 509–517.

C. Yap, C. Chua, Z. Dong, Z. Liu, D. Zhang, L. Loh, et al., Review of selective laser melting: materials and applications, Appl. Phys. Rev. 2 (2015), 041101.

D. Gu, Laser Additive Manufacturing (AM): Classification, Processing Philosophy,and Metallurgical Mechanisms. Laser Additive Manufacturing of High-Performance Materials, Springer, 2015 1571.

S.A. Khairallah, A.T. Anderson, A. Rubenchik,W.E. King, Laser powder-bed fusion additive manufacturing: physics of complex melt flow and formation mechanisms of pores, spatter, and denudation zones, Acta Mater. 108 (2016) 3645.

Thijs, L, Karolien Kempen, Jean-Pierre Kruth, Jan Van Humbeeck. Fine-structured aluminium products with controllable texture by selective laser melting of pre-alloyed AlSi10Mg powder. Acta Materialia 61 (2013) 1809–1819.

Andrew Townsend, Radu Racasan, Richard Leach, Nicola Senin, Adam Thompson, Andrew Ramsey, David Bate, Peter Woolliams, Stephen Brown, Liam Blunt, An interlaboratory comparison of X-ray computed tomography measurement for texture and dimensional characterisation of additively manufactured parts (2018) Additive Manufacturing, 23, pp. 422-432.

 Reviewer #2:

Comment 1: Each one of the cited references within the body of the paper should be discussed individually and explicitly to demonstrate their significance to the study. Also note that cited authors' surname should be used as the subject of a verb, and then state in one or two sentences what they claim, what evidence they provide to support their claim, and how the work is evaluated. Introduction is expected to have an extensive literature review followed by an in-depth and critical analysis of the state of the art. References section should be extensive about information connecting with addictive methods, materials and structures. I suggest add information to better describe what other researchers have done in this area, for example research prof Richard Leach or prof Pero Raos. I suggest add important and new articles from this field:

Experimental analysis of surface roughness and surface texture of machined and fused deposition modelled parts. (2014) Tehnicki Vjesnik, 21 (1), pp. 217-221.

Dimensional accuracy of camera casing models 3D printed on Mcor IRIS: A case study (2016) Advances in Production Engineering And Management, 11 (4), pp. 324-332.

An interlaboratory comparison of X-ray computed tomography measurement for texture and dimensional characterisation of additively manufactured parts (2018) Additive Manufacturing, 23, pp. 422-432.

 Response: Thanks for the suggestions. We have made the corresponding revisions in the revised manuscript (please see Table 1). The suggested references by the Reviewer #2 were added to the revised manuscript:

Andrew Townsend, Radu Racasan, Richard Leach, Nicola Senin, Adam Thompson, Andrew Ramsey, David Bate, Peter Woolliams, Stephen Brown, Liam Blunt, An interlaboratory comparison of X-ray computed tomography measurement for texture and dimensional characterisation of additively manufactured parts (2018) Additive Manufacturing, 23, pp. 422-432.

X.P. Li , G. Ji , Z. Chen, A. Addad , Y. Wu , H.W. Wang, J. Vleugels, J. Van Humbeeck, J.P. Kruth, Selective laser melting of nano-TiB2 decorated AlSi10Mg alloy with high fracture strength and ductility .Acta Materialia 129 (2017) 183-193.

Bandar AlMangour, Dariusz Grzesiak, Tushar Borkar, Jenn-Ming Yang. Densification behavior, microstructural evolution, and mechanical properties of TiC/316L stainless steel nanocomposites fabricated by selective laser melting, Materials and Design 138 (2018) 119–128.

Bandar AlMangoura, Dariusz Grzesiak , Jinquan Cheng , Yavuz Ertas, Thermal behavior of the molten pool, microstructural evolution, and tribological performance during selective laser melting of TiC/316L stainless steel nanocomposites: Experimental and simulation methods. Journal of Materials Processing Tech. 257 (2018) 288–301.

I. Rosenthal, A. Stern, N. Frage, Strain rate sensitivity and fracture mechanism of AlSi10Mg parts produced by selective laser melting. Materials Science & Engineering A 682 (2017) 509–517.

C. Yap, C. Chua, Z. Dong, Z. Liu, D. Zhang, L. Loh, et al., Review of selective laser melting: materials and applications, Appl. Phys. Rev. 2 (2015), 041101.

D. Gu, Laser Additive Manufacturing (AM): Classification, Processing Philosophy,and Metallurgical Mechanisms. Laser Additive Manufacturing of High-Performance Materials, Springer, 2015 1571.

S.A. Khairallah, A.T. Anderson, A. Rubenchik, W.E. King, Laser powder-bed fusion additive manufacturing: physics of complex melt flow and formation mechanisms of pores, spatter, and denudation zones, Acta Mater. 108 (2016) 3645.

Thijs, L, Karolien Kempen, Jean-Pierre Kruth, Jan Van Humbeeck. Fine-structured aluminium products with controllable texture by selective laser melting of pre-alloyed AlSi10Mg powder. Acta Materialia 61 (2013) 1809–1819.

Reviewer 2 Report

The paper topic is interesting and paper can be published but only after the above suggestions:

Introduction: Each one of the cited references within the body of the paper should be discussed individually and explicitly to demonstrate their significance to the study. Also note that cited authors' surname should be used as the subject of a verb, and then state in one or two sentences what they claim, what evidence they provide to support their claim, and how the work is evaluated. Introduction is expected to have an extensive literature review followed by an in-depth and critical analysis of the state of the art. References section should be extensive about information connecting with addictive methods, materials and structures. I suggest add information to better describe what other researchers have done in this area, for example research prof Richard Leach or prof Pero Raos. I suggest add important and new articles from this field:

Experimental analysis of surface roughness and surface texture of machined and fused deposition modelled parts. (2014) Tehnicki Vjesnik, 21 (1), pp. 217-221.

Dimensional accuracy of camera casing models 3D printed on Mcor IRIS: A case study (2016) Advances in Production Engineering And Management, 11 (4), pp. 324-332.

An interlaboratory comparison of X-ray computed tomography measurement for texture and dimensional characterisation of additively manufactured parts (2018) Additive Manufacturing, 23, pp. 422-432.

The discussion is shallow and needs more details, the observations and future trends. This chapter should be connected with others published papers.

Some of the bullet points on the conclusion are simplistic;  Please try to emphasize your novelty, put some quantifications, and comment on the limitations. This is a very common way to write conclusions for a learned academic journal. The conclusions should highlight the novelty and advance in understanding presented in the work.

Author Response

Reviewer #2:

Comment 1: Each one of the cited references within the body of the paper should be discussed individually and explicitly to demonstrate their significance to the study. Also note that cited authors' surname should be used as the subject of a verb, and then state in one or two sentences what they claim, what evidence they provide to support their claim, and how the work is evaluated. Introduction is expected to have an extensive literature review followed by an in-depth and critical analysis of the state of the art. References section should be extensive about information connecting with addictive methods, materials and structures. I suggest add information to better describe what other researchers have done in this area, for example research prof Richard Leach or prof Pero Raos. I suggest add important and new articles from this field:

Experimental analysis of surface roughness and surface texture of machined and fused deposition modelled parts. (2014) Tehnicki Vjesnik, 21 (1), pp. 217-221.

Dimensional accuracy of camera casing models 3D printed on Mcor IRIS: A case study (2016) Advances in Production Engineering And Management, 11 (4), pp. 324-332.

An interlaboratory comparison of X-ray computed tomography measurement for texture and dimensional characterisation of additively manufactured parts (2018) Additive Manufacturing, 23, pp. 422-432.

 Response: Thanks for the suggestions. We have made the corresponding revisions in the revised manuscript (please see Table 1). The suggested references by the Reviewer #2 were added to the revised manuscript:

Andrew Townsend, Radu Racasan, Richard Leach, Nicola Senin, Adam Thompson, Andrew Ramsey, David Bate, Peter Woolliams, Stephen Brown, Liam Blunt, An interlaboratory comparison of X-ray computed tomography measurement for texture and dimensional characterisation of additively manufactured parts (2018) Additive Manufacturing, 23, pp. 422-432.

X.P. Li , G. Ji , Z. Chen, A. Addad , Y. Wu , H.W. Wang, J. Vleugels, J. Van Humbeeck, J.P. Kruth, Selective laser melting of nano-TiB2 decorated AlSi10Mg alloy with high fracture strength and ductility .Acta Materialia 129 (2017) 183-193.

Bandar AlMangour, Dariusz Grzesiak, Tushar Borkar, Jenn-Ming Yang. Densification behavior, microstructural evolution, and mechanical properties of TiC/316L stainless steel nanocomposites fabricated by selective laser melting, Materials and Design 138 (2018) 119–128.

Bandar AlMangoura, Dariusz Grzesiak , Jinquan Cheng , Yavuz Ertas, Thermal behavior of the molten pool, microstructural evolution, and tribological performance during selective laser melting of TiC/316L stainless steel nanocomposites: Experimental and simulation methods. Journal of Materials Processing Tech. 257 (2018) 288–301.

I. Rosenthal, A. Stern, N. Frage, Strain rate sensitivity and fracture mechanism of AlSi10Mg parts produced by selective laser melting. Materials Science & Engineering A 682 (2017) 509–517.

C. Yap, C. Chua, Z. Dong, Z. Liu, D. Zhang, L. Loh, et al., Review of selective laser melting: materials and applications, Appl. Phys. Rev. 2 (2015), 041101.

D. Gu, Laser Additive Manufacturing (AM): Classification, Processing Philosophy,and Metallurgical Mechanisms. Laser Additive Manufacturing of High-Performance Materials, Springer, 2015 1571.

S.A. Khairallah, A.T. Anderson, A. Rubenchik, W.E. King, Laser powder-bed fusion additive manufacturing: physics of complex melt flow and formation mechanisms of pores, spatter, and denudation zones, Acta Mater. 108 (2016) 3645.

Thijs, L, Karolien Kempen, Jean-Pierre Kruth, Jan Van Humbeeck. Fine-structured aluminium products with controllable texture by selective laser melting of pre-alloyed AlSi10Mg powder. Acta Materialia 61 (2013) 1809–1819.

 Comment 2: The discussion is shallow and needs more details, the observations and future trends. This chapter should be connected with others published papers. Some of the bullet points on the conclusion are simplistic; Please try to emphasize your novelty, put some quantifications, and comment on the limitations. This is a very common way to write conclusions for a learned academic journal. The conclusions should highlight the novelty and advance in understanding presented in the work.

 Response: Thanks for the suggestion. We have revised the corresponding parts and the last two paragraphs in the Results and Discussions were reorganized. Please see below Table 1 for the changes made.

Round  2

Reviewer 1 Report

Well-done. Just minor comment: There is a typo error that i have noticed in Reference 6. The author name is AlMangour and NOT AlMangoura. Please check the rest of the references.

Reviewer 2 Report

Paper is ready for publication